# English Classroom Anxiety, Learning Style and English Achievement in Chinese University EFL Students

Meihua Liu 

Department of Foreign Languages and Literatures, Tsinghua University, Beijing 100084, China;
liumeihua@mail.tsinghua.edu.cn

**Abstract:** The present study examined the relationship between English classroom anxiety (ECA) and learning style and their predictive effects on students' English achievement. In total, 691 Chinese university EFL (English as a foreign language) students answered the English Classroom Anxiety Scale, the Perceptual Learning Styles Preferences Questionnaire and a background information questionnaire. Major findings were: (a) the two scales were highly reliable and significantly inversely related to each other, (b) the respondents generally had a medium ECA level and selected auditory, kinesthetic, visual, tactile, group and individual styles as their minor preferences, (c) no significant differences occurred in ECA levels between students of varying genders and disciplines, (d) male students preferred group learning significantly more and individual learning significantly less than their female peers, and engineering students preferred group learning significantly more and individual learning significantly less than their peers of social sciences and humanities, (e) ECA was significantly negatively correlated with and predicted students' English achievement, and (f) each learning style was significantly positively correlated with students' English achievement, and visual and group styles significantly positively predicated the latter. These findings confirm the role of foreign language anxiety and learning style in second/foreign language learning.

**Keywords:** English classroom anxiety; learning style; gender difference; discipline difference; predictive effect

## 1. Introduction

As an important affective and psychological factor, foreign language anxiety (FLA) has been much examined in second/foreign language (SL/FL) learning [1–5]. Research shows that it is not only consistently negatively related to SL/FL learning outcomes measured in varying forms but also interacts with many other linguistic, educational, psychological, individual and affective factors, as reviewed below [6–9]. Mixed findings are often found about FLA and two important individual factors—gender and discipline, which might be due to different samples sizes and methods used in the studies (e.g., [3,10–14]).

Every individual learns in his or her own way, which is often linked to past experiences, characters, the learning environment and so on, both emotionally and physically. To better understand how learners learn an SL/FL and how it is related to SL/FL learning outcomes, learning style has been increasingly researched in SL/FL learning in recent decades (e.g., [15–17]). Studies reveal mixed findings about the relationship between learning styles and SL/FL learning outcomes (e.g., [18,19]). Studies also show that ESL/EFL (English as an SL/FL) students generally prefer kinesthetic and tactile learning styles the most and group learning the least (e.g., [20,21]). Nevertheless, differences do exist among students of different genders, disciplines, cultures, ages, education levels and learning environments [16,22–24]. Though learning styles are assumed to interact with the environment and other variables like strategy use (e.g., [16,25]), little research can be found on the relationship between foreign language anxiety and learning styles.

All these findings imply that more and continuous research is needed on both FLA and learning styles in SL/FL learning. Moreover, as discussed in the socio-interactionist theory, learning happens through the continuous interaction of endogenous and exogenous factors [26]. SL/FL learning is highly likely to be affected by students' learning style preferences and foreign language anxiety. Yet, the relationship between the two and their effects on SL/FL learning outcomes have hardly been researched. Given that COVID-19 has posed many damages to human health and formal learning in recent years, the present research aimed to investigate the relationship between English classroom anxiety and learning style and their predictive effects on Chinese university students' English achievement during COVID-19. To be more specific, the study sought to examine the profiles of and differences in English classroom anxiety and learning styles between students of differing genders and disciplines, the relationship between English classroom anxiety and learning styles, and the predictive effects of the two variables on students' English achievement. By achieving these objectives, the current study makes one of the few contributions on the relationship between English classroom anxiety and learning styles and their predictive effects for SL/FL learning outcomes. As such, the study enriches the current literature and the results better our understanding of the two variables and their roles in SL/FL learning.

## 2. Literature Review

### 2.1. Learning Styles

It is widely acknowledged that SL/FL learning is deeply influenced by learners' internal factors, such as motivation, attitudes, self-confidence, anxiety, personality and learning styles [27]. Of these internal factors, many have been extensively researched, while research on learning styles in relation to SL/FL learning is still inadequate despite an increasing attention to learning styles since the 1970s [28].

Keefe [29] (p. 4) defined learning styles as "characteristic cognitive, affective, and physiological behaviors that serve as relatively stable indicators of how learners perceive, interact with, and respond to the learning environment". According to Reid [30], the learning style is a student's natural, habitual, and preferred way of absorbing, processing and retaining new knowledge. Clearly, learning styles are combinations of an individual's cognitive, affective and psychological characteristics that interact with the environment [31]. Since individuals may prefer various different ways of learning, the learning style consists of different components. Dunn et al. [32] categorized learning styles as visual, tactile and kinesthetic. Reid [20] classified learning styles into six types: auditory (preferring to learn through oral–aural channel), visual (preferring to learn through seeing/visual channel), kinesthetic (preferring to learn through experiential learning/total physical involvement), tactile (preferring to learn through hands-on activities/doing lab experiments), individual learning (preferring to learn through working alone) and group learning (preferring to learn through working with others and participating in group works). To measure learning styles of non-native speakers of English, Reid [20] developed the 30-item five-point-Likert Perceptual Learning Style Preference Questionnaire (PLSPQ).

Although Pelegrín's [33] review shows that the PLSPQ is an instrument in need of profound improvement, it is the most recent and widely used instrument for ESL/EFL learners (e.g., [15–22,24,25,33–38]). These studies show that ESL/EFL students generally prefer kinesthetic and tactile learning styles the most and prefer group learning the least. For example, Reid [20] researched learning styles on about 1300 ESL students studying in America and found that ESL students significantly differed from native speakers of English in their perceptual and social learning style preferences. Most ESL students preferred kinesthetic and tactile learning while native speakers of English were less tactile than all ESL students and less kinesthetic than Korean, Arabic, Chinese and Spanish speakers. Both groups preferred group learning the least. These findings were generally supported by Rossi-Le's [21] study of 147 adult immigrant ESL students in the US with backgrounds of Chinese, Laotian, Vietnamese, Spanish, Cambodian, Japanese, Polish and Korean. Gao's [16] study of 250 second-year non-English majors from a Chinese university indicated that almost all the

participants were multi-style learners while preferring tactile, visual and kinesthetic styles the most and group style the least. Alkahtani's [22] research of 667 EFL students studying at Yanbu English Language Institute revealed that the overall dominant perceptual style preferences were auditory and group learning. Akbarian et al.'s [15] investigation of 235 Iranian tertiary EFL learners showed that kinesthetic, auditory, visual and tactile modalities were the participants' major learning styles while individual and group styles were their minor styles and that group learning was the least preferred style. Ha's [18] research of 162 students at UNETI, Vietnam revealed that group learning was the participants' most preferred style. These studies imply that differences in learning styles do exist as the learner population and learning context are different.

Differences also exist in learning styles in students of different genders and disciplines [23,39,40]. For example, in addition to the finding that the participants mostly preferred kinesthetic and auditory styles and liked individual and group styles the least, Peacock's [40] study of 206 Hong Kong university EFL students revealed that humanities students chose the auditory style as their major preference and the individual style as their minor preference and had a negative opinion of group learning, while science students chose the auditory style as their minor preference and held a negative opinion of individual learning. Al Khatib and Ghosheh's [23] study of 210 students of Al Ain University of Science and Technology showed that education students were more tactile learners than those in other fields of study, while law students were more group learners and pharmacy students were more individual learners than those in other fields. They also found that male students were more auditory and tactile learners while female students were more group learners. Hyland's [39] study of 440 Japanese university students indicated that the participants had no major style and that female students demonstrated stronger preferences for all styles than males. Mozayan et al.'s [36] study of 107 Iranian medical sciences students also found that female students demonstrated stronger preferences for the styles than their male peers, which was supported by the finding in Akbarian et al. [15]. Nevertheless, no significant gender difference in learning styles was found in Zokaee et al.'s [25] study of 54 EFL learners at Tarbiat Moallem University.

Meanwhile, though not many studies have examined the relationship between learning styles and students' achievements in SL/FL learning, some interesting findings have been revealed (e.g., [15,18]). For example, Akbarian et al. [15] found that the participants' tactile style scores significantly correlated with vocabulary knowledge. Ha [18] found a significant relationship between the students' learning styles and their English language proficiency. Nosratinia and Soleimannejad's [17] study of 595 undergraduate EFL learners aged 18 to 25 indicated that significant and positive relationships existed between the participants' critical thinking and total score of perceptual learning styles and that tactile learning style preference was the best predictor of EFL learners' critical thinking. Malsawmkimi and Fanai's [19] study of 192 secondary school EFL students found no correlation between students' academic achievements and the scores on different learning styles.

In addition, learning styles have been found to interact with learning strategies, though limited research is available on the interaction of learning styles with other variables in SL/FL learning [16,22]. Gao [16] showed that there were complex relationships between learning styles and learning strategy preferences, as found in Zokaee et al. [25]. Alkahtani's [22] research found significant correlations between perceptual language learning styles and language learning strategy use.

*2.2. English Classroom Anxiety*

As a situation-specific type of anxiety, foreign language anxiety (FLA) has been extensively researched since the 1970s (e.g., [2–5]). It refers to "the worry and negative emotional reaction aroused when learning or using a second language" [41] (p. 27). This is because SL/FL learners often become doubtful and stressed when learning/using an SL/FL due to uncertain or unknown linguistic and socio-cultural rules [1]. Research on FLA shows

that it may lead to deficits in SL/FL learning and performance and low self-confidence in SL/FL learners [1,2].

Anxiety exists in almost every aspect of SL/FL learning (i.e., listening, speaking, reading, and writing) [1]. Of various types of FLA, foreign language classroom anxiety has caught much attention, probably because of its close relation to formal classroom teaching and learning [5]. Foreign language classroom anxiety (FLCA) is "a distinct complex of self-perceptions, beliefs, feelings and behaviors related to classroom learning arising from the uniqueness of the language learning process" [1] (p. 128). To measure this anxiety, Horwitz et al. [1] developed the 33-item 5-point Likert Foreign Language Classroom Anxiety Scale (FLCAS), which has then been widely adopted or adapted to measure learners' anxiety in language classrooms or speaking anxiety in varying SL/FL contexts (e.g., [2,4–8,12,42,43]). Believing that situational anxiety affects not only formal language training but also informal language experiences, Gardner [2] developed the 8-item French Classroom Anxiety Scale (FCAS) to measure anxiety in French language classrooms and found that anxiety was negatively related to L2 motivation and French learning outcomes.

These studies, using either of the scales, together with those using other methods [4,8,9,13,44,45], reveal that FLCA is consistently negatively related to learners' language performance, and that FLCA can be attributed to various learner-related, teacher-related and contextual variables, such as low language proficiency, low self-confidence, past learning experiences, fear of losing face, personality and pursuit of perfection. For example, Zhang's [5] meta-analysis analyzed 55 independent samples with more than 10,000 participants and found that FLA was significantly negatively correlated with FL performance ($r = -0.34$, $p < 0.01$), which remained stable across groups with different FL proficiency levels. These results were generally supported by Dikmen's [13] meta-analysis of 69 studies from fourteen countries.

In regard to gender difference, some studies report that female learners experience higher levels of FLA than their male peers (e.g., [11,12,46,47]). For example, Dewaele's [12] study of 1287 female and 449 male participants revealed that female students demonstrated both more foreign language enjoyment and more foreign language classroom anxiety than their male peers. The researcher believed that this was because female learners were more emotionally involved in the FL learning and hence experiencing more emotional highs and lows than their male peers. Conversely, some studies show that male students are more anxious in FL classrooms (e.g., [48]). In a study of 64 Indonesian EFL learners, Hasan and Fatimah [48] discovered that male students exhibited more anxiety than their female peers. Contrary to this, some studies reveal no significant gender differences in FLA (e.g., [49,50]).

Likewise, mixed findings have been found about foreign language anxiety and disciplines. Many studies find that there are significant differences in FLA among learners with varying disciplinary backgrounds (e.g., [3,10,14,51]). For example, Kimura's [51] study of 452 Japanese freshmen showed that math students experienced more anxiety than social science students. Torudom and Taylor [14] found that science students displayed a higher level of reading anxiety than non-science students. Altunel [10] found that students of natural sciences experienced more fear of negative emotions and a relatively higher level of FLA. On the other hand, Rajab et al. [52] found no significant differences in FLA among students of different disciplines, which might be because their studies involved no more than 100 participants.

Meanwhile, these studies show that FLCA interacts with many other variables, including age, gender, motivation, strategy use, field of study and foreign language enjoyment. For example, Liu's [3] study of 934 Chinese first-year university students revealed significant differences in FLCA and its effects on English achievement between male and female students and between those from different disciplines. Nevertheless, little research is available on the relationship between FLA and learning styles.

*2.3. The COVID-19 Context*

Four years has passed since the outbreak of COVID-19 in late 2019. Because of its fast spreading speed and varying degrees of damage to human health, many schools in the world, including those in China, shifted online when COVID-19 was serious and then hybrid teaching and learning when it was not so serious, to prevent fast infection and reduce loss. When the present study was conducted in the spring semester of 2021 in China, there were occasionally tens of COVID-19 cases reported in certain areas of a province. People were generally free to travel but encouraged to wear masks in public places, and no gatherings of more than eight were allowed. Students normally took classes in real classrooms but were required to take them online if they had just returned from a place with COVID-19 cases or if they had a fever.

Though development in information and communications technology eliminates the barriers of time, space and pace, and creates greater flexibility in learning and teaching, students may experience isolation, distraction and frustration due to the lack of adequate face to-face interaction and instability of the internet [53]. They thus may have a reduced sense of belonging, lowered willingness and motivation to study and heightening anxiety in learning and using the language [53]. For example, though van der Velde et al. [54] found that COVID-19 did not negatively affect FL learners' performance, Liu and Yuan's [53] longitudinal quantitative study showed that university students had high levels of English language classroom anxiety (ELCA) and listening anxiety throughout the semester. They thus claimed that ELCA and listening anxiety were serious issues in the pandemic FL learning context. Nevertheless, similar studies are rarely found from the period during COVID-19, which motivated the present study.

**3. Research Questions**

According to the socio-interactionist theory [55], a learner's cognitive development and learning ability are guided and mediated by their interactions with internal and external factors. Of these factors, learning style and FLA have demonstrated their respective roles in SL/FL learning, as reviewed above. Nevertheless, little research can be found on the relationship between FLA and learning styles and their effects on SL/FL learning. In addition, research shows that both FLA and learning styles interact with many other linguistic, educational, psychological, individual and affective factors (e.g., [7–9,16,25]). And mixed findings have been found in both FLA and learning styles among students of varying genders, disciplines and even SL/FL learning outcomes (e.g., [3,10–13,22,23]). All these attest to the importance of more and continuous research on FLA and learning styles in SL/FL learning. This is especially true of the COVID-19 period, in which a lot of damage was done to human health and formal learning, though little research has been conducted in terms of SL/FL learning in this context [53,54]. Hence, the present research aimed to investigate the relationship between English classroom anxiety and learning styles and their predictive effects on Chinese university students' English achievement during COVID-19. The following questions were of particular interest:

(1) What are the profiles of and differences in English classroom anxiety and learning styles?
(2) How is English classroom anxiety related to learning styles?
(3) How do English classroom anxiety and learning styles predict students' English achievement?

**4. Methodology**

*4.1. Participants*

With a mean age of 20.17 (SD = 1.56) and an age range of 18 to 30, 691 (297 male and 394 female) university students participated in the present study. The participants were studying in different years of college, and they came from three major disciplines: engineering (412/59.6%), natural sciences (22/3.2%) and social sciences and humanities

(257/37.2%). They reported spending a mean of 1.58 h (SD = 0.716) using English (i.e., listening, speaking, reading and writing) per day.

*4.2. Instruments*

English Classroom Anxiety Scale. This 8-item English Classroom Anxiety Scale (ECAS) (Cronbach alpha *a* = 0.751) was adapted from Gardner [2] to measure students' anxiety in English language classrooms. The word 'French' was changed to 'English' in all items to suit the present study.

Perceptual Learning Styles Preferences Questionnaire. Reid's [20] 30-item Perceptual Learning Styles Preferences Questionnaire (PLSPQ) (*a* = 0.883) was used in the present study, which covers 6 components with each component having 5 items. To analyze the data received from the PLSPQ, Reid [20] provided 3 cut-off scores for major learning style preference (38–50), minor learning style preference (25–37), and negligible learning style preference (24 or less).

All the ECAS and PLSPQ items were placed on a 5-point Likert scale, ranging from 'strongly agree' to 'strongly disagree' with values of 1–5 assigned to the descriptors, respectively.

Background Information Questionnaire. The Background Information Questionnaire aimed to collect information about the participants such as age, gender, major area of study and year of study.

English achievement. The participants' English achievement was measured based on self-rated proficiency in overall English on a scale of 1 (very poor) to 10 (native-like).

*4.3. Data Collection Procedure and Analysis*

This study was conducted in the first semester of 2022 when COVID-19 was severe in China. After the study was approved by the Research Committee of the department, all the questionnaires, together with a consent form, were distributed online to university students nationwide. All the participation was voluntary. Finally, 691 valid questionnaires were collected, which were then analyzed with SPSS version 22. Means and standard deviations were computed to examine ECAS levels and learning style preferences; t-tests were run to explore differences between genders and disciplines; correlation analyses were run to reveal relationships among the measured variables; and regression analyses were conducted to investigate whether and how ECAS and learning styles predicted students' English achievement.

## 5. Results

*5.1. English Classroom Anxiety, Learning Style Patterns and Correlations of the Whole Sample*

As shown in Table 1, the respondents scored 5.54 on SOEP and 2.92 on ECAS, meaning that they were generally intermediate learners of English and had a medium level of English classroom anxiety. They scored 35.22, 35.53, 34.38, 35.54, 33.16 and 33.92 on auditory, kinesthetic, visual, tactile, group and individual styles, respectively, indicating that all the learning styles were the respondents' minor style preferences. Of all the six learning styles, the students preferred the tactile style the most, followed by kinesthetic, auditory, visual, individual and group styles, respectively.

Meanwhile, the six learning styles were significantly positively related to SOEP ($r = 0.077 \sim 0.213$, $p \leq 0.05$) but inversely related to ECAS ($r = -0.119 \sim -0.281$, $p \leq 0.001$). This is, a learner who preferred the auditory, kinesthetic, visual, tactile, group or individual style more tended to self-rate his/her overall English proficiency higher and be less anxious in English classrooms, or vice versa. Moreover, SOEP was significantly negatively related to ECAS ($r = -0.288$, $p \leq 0.001$), indicating that a respondent who self-rated his/her overall English proficiency higher tended to be less anxious in English classrooms.

**Table 1.** Means, SDs and correlations of the measures of the whole sample (N = 691).

| | Mean | SD | 2 | 3 | 4 | 5 | 6 | 7 | 8 |
|---|---|---|---|---|---|---|---|---|---|
| 1 SOEP | 5.54 | 3.12 | −0.288 ** | 0.168 ** | 0.166 ** | 0.213 ** | 0.124 ** | 0.159 ** | 0.077 * |
| 2 ECAS | 2.92 | 0.72 | 1 | −0.202 ** | −0.281 ** | −0.216 ** | −0.181 ** | −0.200 ** | −0.119 ** |
| 3 Auditory | 35.22 | 6.35 | | 1 | 0.661 ** | 0.514 ** | 0.557 ** | 0.482 ** | 0.126 ** |
| 4 Kinesthetic | 35.53 | 6.87 | | | 1 | 0.512 ** | 0.674 ** | 0.547 ** | 0.090 * |
| 5 Visual | 34.38 | 6.56 | | | | 1 | 0.556 ** | 0.407 ** | 0.314 ** |
| 6 Tactile | 35.54 | 6.51 | | | | | 1 | 0.481 ** | 0.187 ** |
| 7 Group | 33.16 | 8.01 | | | | | | 1 | −0.168 ** |
| 8 Individual | 33.92 | 7.91 | | | | | | | 1 |

Notes. ** = $p \leq 0.01$; * = $p \leq 0.05$; SOEP = self-rated overall English proficiency; ECAS = English Classroom Anxiety Scale.

### 5.2. English Classroom Anxiety and Learning Styles: Gender Difference

As shown in Table 2, male students scored 5.45 on SOEP, 2.87 on ECAS and 33.10 to 35.54 on auditory, kinesthetic, visual, tactile, group and individual styles, respectively; female students scored 5.61 on SOPE, 2.96 on ECAS and 32.55 to 35.81 on the six learning styles, respectively. These results show that both male and female students were generally intermediate learners of English, had a medium English classroom anxiety level and selected auditory, kinesthetic, visual, tactile, group and individual styles as their minor preferences. Of all the six learning styles, male students preferred the kinesthetic style the most, followed by auditory, tactile, visual, group and individual styles, respectively; female students preferred the tactile style the most, followed by kinesthetic, auditory, individual, visual and group styles, respectively.

**Table 2.** Means, SDs and *t*-test results of the measured variables between genders.

| Measures | Male (N = 297) | | Female (N = 394) | | *t*-Test Results | | |
|---|---|---|---|---|---|---|---|
| | Mean | SD | Mean | SD | t | p | Cohen's d |
| SOEP | 5.45 | 2.16 | 5.61 | 3.69 | −0.700 | 0.484 | / |
| ECAS | 2.87 | 0.725 | 2.96 | 0.717 | −1.662 | 0.097 | / |
| Auditory | 35.31 | 6.567 | 35.16 | 6.198 | 0.319 | 0.750 | / |
| Kinesthetic | 35.54 | 6.953 | 35.53 | 6.821 | 0.014 | 0.989 | / |
| Visual | 34.32 | 7.152 | 34.43 | 6.082 | −0.218 | 0.828 | / |
| Tactile | 35.19 | 6.861 | 35.81 | 6.232 | −1.253 | 0.210 | / |
| Group | 33.98 | 8.175 | 32.55 | 7.836 | 2.334 * | 0.020 | 0.18 |
| Individual | 33.19 | 8.371 | 34.47 | 7.508 | −2.111 * | 0.035 | 0.16 |

Notes. * = $p \leq 0.05$; effect size of Cohen's d: small = d $\leq$ 0.2; medium = d = 0.5; large = d $\geq$ 0.8 [56].

Concurrently, *t*-test results show that male students differed significantly from female students in group (t = 2.334, $p \leq 0.05$, d = 0.18) and individual (t = −2.111, $p \leq 0.05$, d = 0.16) styles, with a medium effect size. Namely, male students preferred group learning significantly more and individual learning significantly less than their female peers.

### 5.3. English Classroom Anxiety and Learning Styles: Discipline Difference

Since only 22 students of natural sciences participated in the present study, only differences between students of engineering and those of social sciences and humanities (SSH) were explored. The results are reported in Table 3, which shows that engineering students scored 5.40 on SOEP, 2.93 on ECAS and 33.31 to 35.83 on auditory, kinesthetic, visual, tactile, group and individual styles, respectively, while SSH students scored 5.73 on SOEP, 2.89 on ECAS and 31.88 to 35.595 on the six learning styles, respectively. These results show that both engineering and SSH students were generally intermediate learners of English, had a medium English classroom anxiety level and selected auditory, kinesthetic, visual, tactile, group and individual styles as their minor preferences. Of all the six learning styles, engineering students preferred the tactile style the most, followed by kinesthetic,

auditory, visual, group and individual styles, respectively; SSH students preferred the kinesthetic style the most, followed by tactile, auditory, individual, visual and group styles, respectively.

**Table 3.** Means, SDs and *t*-test results of the measures between disciplines.

| Measures | Engineering (N = 412) | | SSH (Social Sciences and Humanities) (N = 257) | | *t*-Test Results | | |
|---|---|---|---|---|---|---|---|
| | Mean | SD | Mean | SD | t | p | Cohen's d |
| SOEP | 5.40 | 2.214 | 5.73 | 4.249 | −1.290 | 0.197 | / |
| ECAS | 2.93 | 0.742 | 2.89 | 0.693 | 0.667 | 0.505 | / |
| Auditory | 35.47 | 6.507 | 34.96 | 6.000 | 1.013 | 0.311 | / |
| Kinesthetic | 35.63 | 0.709 | 35.595 | 6.470 | 0.066 | 0.948 | / |
| Visual | 34.48 | 6.858 | 34.33 | 6.064 | 0.280 | 0.780 | / |
| Tactile | 35.83 | 6.555 | 35.26 | 6.401 | 1.093 | 0.275 | / |
| Group | 34.09 | 8.032 | 31.88 | 7.720 | 3.504 ** | 0.000 | 0.28 |
| Individual | 33.31 | 8.060 | 34.62 | 7.578 | −2.103 * | 0.036 | 0.17 |

Notes. ** = $p \leq 0.01$; * = $p \leq 0.05$.

Concurrently, *t*-test results show that engineering students differed significantly from SSH students in group (t = 3.504, $p \leq 0.001$, d = 0.28) and individual (t = −2.103, $p \leq 0.05$, d = 0.17) styles, with a medium effect size. That is, engineering students preferred group learning significantly more and individual learning significantly less than their SSH peers.

*5.4. Correlations among Measured Variables*

As shown in Table 4, similar patterns of correlations as those for the whole sample presented in Table 2 occurred for male, female, engineering and SSH students except that SSH students' SOEP was generally not significantly related to learning styles. That is, for all the four specific samples, the six learning styles were significantly positively related to SOEP and negatively to ECAS, and SOEP was significantly negatively related to ECAS.

**Table 4.** Correlations among measured variables.

| | | ECAS | Auditory | Kinesthetic | Visual | Tactile | Group | Individual |
|---|---|---|---|---|---|---|---|---|
| Male (N = 297) | SOEP | −0.399 ** | 0.261 ** | 0.303 ** | 0.312 ** | 0.251 ** | 0.298 ** | 0.099 |
| | ECAS | 1 | −0.197 ** | −0.297 ** | −0.232 ** | −0.185 ** | −0.218 ** | −0.081 |
| Female (N = 394) | SOEP | −0.255 ** | 0.134 ** | 0.112 * | 0.179 ** | 0.069 | 0.108 * | 0.068 |
| | ECAS | 1 | −0.205 ** | −0.270 ** | −0.204 ** | −0.184 ** | −0.178 ** | −0.161 ** |
| Engineering (N = 412) | SOEP | −0.402 ** | 0.277 ** | 0.293 ** | 0.283 ** | 0.231 ** | 0.286 ** | 0.065 |
| | ECAS | 1 | −0.205 ** | −0.339 ** | −0.221 ** | −0.214 ** | −0.258 ** | −0.078 |
| SSH (N = 257) | SOEP | −0.207 ** | 0.092 | 0.064 | 0.167 ** | 0.052 | 0.076 | 0.084 |
| | ECAS | 1 | −0.210 ** | −0.168 ** | −0.176 ** | −0.143 * | −0.097 | −0.204 ** |

Notes. * $p \leq 0.05$; ** $p \leq 0.01$; SSH = social sciences and humanities.

*5.5. Predictors for English Achievement*

To explore whether ECAS and learning styles predicted students' English achievement, multiple stepwise regression analyses were run for the whole sample as well as for male, female, engineering and SSH students, respectively, with students' self-rated overall English proficiency (SOEP) as the dependent variable and ECAS and the six learning styles as independent variables. The results are reported in Table 5, which shows that ECAS (English classroom anxiety scale) (β = −0.254, t = −6.874, $f^2$ = 0.09) and the visual style (β = 0.158, t = 4.286, $f^2$ = 0.12) significantly predicated the whole sample's SOEP, accounting for 8.3% and 2.4% of the total variance, respectively. ECAS (β = −0.329, t = −6.175, $f^2$ = 0.19), visual (β = 0.164, t = 2.746, $f^2$ = 0.27) and group (β = 0.146, t = 2.458, $f^2$ = 0.29) styles significantly

predicated male students' SOEP, accounting for 15.9%, 5.1% and 1.6% of the total variance, respectively. ECAS (β = −0.228 t = −4.613, $f^2$ = 0.07) and the visual style (β = 0.133, t = 2.681, $f^2$ = 0.09) were good predictors for female students' SOEP, accounting for 6.5% and 1.7% of the total variance, respectively. ECAS (β = −0.335, t = −7.309, $f^2$ = 0.19), visual (β = 0.149, t = 3.00, $f^2$ = 0.25) and group (β = 0.131, t = 2.614, $f^2$ = 0.27) styles significantly predicated engineering students' SOEP, accounting for 16.1%, 4% and 1.3% of the total variance, respectively. ECAS (β = −0.183, t = −2.967, $f^2$ = 0.04) and the visual style (β = 0.134, t = 2.175, $f^2$ = 0.06) were significant predictors for SSH students' SOEP, accounting for 4.3% and 1.7% of the total variance, respectively.

**Table 5.** Multiple regression coefficients and significance of ECAS and learning style predictors for students' SOEP.

| Predictors | Whole Sample's SOEP | | | | |
| --- | --- | --- | --- | --- | --- |
| | β | t | *p* | VIF | Cohen's $f^2$ |
| ECAS | −0.254 | −6.874 ** | 0.000 | 1.049 | 0.09 |
| Visual | 0.158 | 4.286 ** | 0.000 | 1.049 | 0.12 |

| Predictors | Male students' SOEP | | | | |
| --- | --- | --- | --- | --- | --- |
| | β | t | *p* | VIF | Cohen's $f^2$ |
| ECAS | −0.329 | −6.175 ** | 0.000 | 1.073 | 0.19 |
| Visual | 0.164 | 2.746 ** | 0.006 | 1.346 | 0.27 |
| Group | 0.146 | 2.458 | 0.015 | 1.337 | 0.29 |

| Predictors | Female students' SOEP | | | | |
| --- | --- | --- | --- | --- | --- |
| | β | t | *p* | VIF | Cohen's $f^2$ |
| ECAS | −0.228 | −4.613 ** | 0.000 | 1.044 | 0.07 |
| Visual | 0.133 | 2.681 ** | 0.008 | 1.044 | 0.09 |

| Predictors | Engineering students' SOEP | | | | |
| --- | --- | --- | --- | --- | --- |
| | β | t | *p* | VIF | Cohen's $f^2$ |
| ECAS | −0.335 | −7.309 ** | 0.000 | 1.087 | 0.19 |
| Visual | 0.149 | 3.00 ** | 0.003 | 1.282 | 0.25 |
| Group | 0.131 | 2.614 ** | 0.009 | 1.307 | 0.27 |

| Predictors | SSH students' SOEP | | | | |
| --- | --- | --- | --- | --- | --- |
| | β | t | *p* | VIF | Cohen's $f^2$ |
| ECAS | −0.183 | −2.967 ** | 0.003 | 1.032 | 0.04 |
| Visual | 0.134 | 2.175 * | 0.031 | 1.032 | 0.06 |

Notes. ** = $p \leq 0.01$; * = $p \leq 0.05$; effect size of Cohen's $f^2$: small = $f^2 \leq 0.02$; medium = $f^2$ = 0.15; large = $f^2 \geq 0.35$ [56].

## 6. Discussion

### 6.1. Profiles of and Differences in English Classroom Anxiety and Learning Styles

Statistical analyses showed that the whole sample, as well as male, female, engineering and SSH students, generally had a medium level of English classroom anxiety (ECA), consistent with that in many extant studies (e.g., [3–6,8,12,42,57]). This shows that anxiety still exists in many SL/FL learners even though they have more exposure and access to the target language, further pinpointing the important role of anxiety in SL/FL learning. Nevertheless, this might be because COVID-19 aggravated the respondents' anxiety about English learning, which needs to be further researched.

The present research also revealed that no significant difference occurred in ECA levels between male and female students, as found in Dewaele et al. [49] and Matsuda and Gobel [50]. No significant difference was found between engineering and SSH students either, as found in Rajab et al. [52]. This might be because the participants could choose to go to the classroom or study online due to COVID-19, which somehow offset the differences in ECA levels. This, nevertheless, needs to be validated with more empirical research.

Statistical analyses showed that the whole sample, as well as male, female, engineering and SSH students, generally selected auditory, kinesthetic, visual, tactile, group and individual styles as their minor preferences. Like their Japanese counterparts in Gao [16], the respondents in the present study seemed to be multi-style learners and did not strongly demonstrate any style preference. Different from that in Ha [18] and Koglin et al. [24], they generally preferred tactile and kinesthetic styles the most, and group and individual learning the least, largely consistent with the findings in many similar studies on ESL/EFL students (e.g., [15,16,20,21,36–38]). Because of COVID-19, individual, visual and audio learning have been more often practiced in classrooms and public places like libraries to avoid infection. Nevertheless, this study showed little difference in the participants' style preferences compared with their counterparts in other similar contexts. This suggests that learning style is more a trait than a state characteristic and is not much affected by the context.

Slight differences existed in preferences for learning styles between male and female students, and between engineering and SSH students. Female students preferred the tactile style the most and the group style the least, while male students preferred the kinesthetic style the most and the individual style the least, different from Hyland [39] and Al Khatib and Ghosheh [23]. T-test results showed that male students preferred group learning significantly more and individual learning significantly less than their female peers, different from Zokaee et al. [25]. Meanwhile, engineering students preferred the tactile style the most and the individual style the least, while SSH students preferred the kinesthetic style the most and the group style the least. T-test results revealed that engineering students preferred group learning significantly more and individual learning significantly less than their SSH peers, different from Peacock [40] and Al Khatib and Ghosheh [23]. The findings were contrary to the expectation that male and engineering students might prefer individual learning more while female and SSH students might prefer group learning more, and it is hard to explain that difference, which could have been helped by qualitative data. Nevertheless, all the findings were generally similar to those in the current literature (e.g., [15,23,36,39]), which revealed mixed findings on preferences for learning styles between students of varying genders or disciplines. All these differences further confirm the idea that learning styles differ with age, education level, achievement level, gender, culture and major field (e.g., [15,16,18–20,22,24,33–35,38]).

### 6.2. Predictors for English Proficiency

The present study shows that ECAS was not only significantly negatively correlated with but also inversely predicted students' English achievement, as found in most studies of the current literature (e.g., [3,8,9,13,43–45,53]). These findings further support the negative effect of foreign language anxiety in SL/FL learning regardless of learning environments.

The present study also reveals a significantly positive correlation between each learning style and students' English achievement, as found in Ha [18] and Akbarian et al. [15] though different from Malsawmkimi and Fanai [19]. In addition, visual style significantly positively predicted all samples' English achievement, and group style significantly positively predicted male and engineering students' English achievement. These findings confirm the role of learning styles in SL/FL learning though they need to be further supported with more similar research.

### 7. Conclusions and Implications

The present study examined the relationship between English classroom anxiety and learning styles and their predictive effects on Chinese university students' English achievement. The study revealed the following major findings:

(1)　The English Classroom Anxiety Scale and the Perceptual Learning Styles Preferences Questionnaire were highly reliable and were significantly inversely correlated with each other;

(2)  The respondents generally had a medium level of English classroom anxiety and selected auditory, kinesthetic, visual, tactile, group and individual styles as their minor preferences;

(3)  No significant differences emerged in English classroom anxiety levels between students of different genders and disciplines;

(4)  Male students preferred group learning significantly more and individual learning significantly less than their female peers; engineering students preferred group learning significantly more and individual learning significantly less than their peers of social sciences and humanities;

(5)  English classroom anxiety was significantly negatively related to and predicted students' English achievement;

(6)  Each learning style was significantly positively related to students' English achievement, and visual and group styles significantly positively predicted the latter.

These findings clearly support the importance of FLA and learning styles in SL/FL learning, further supporting the socio-interactionist theory [55]. Hence, both instructors and learners need to be aware of and pay attention to these two issues.

It is important for instructors to create a friendly, mutually supportive and relaxing classroom environment so that students feel at ease and free to learn and use the target language in class, as suggested in Horwitz et al. [1], Liu [3] and Shirvan and Taherian [4]. This is especially important during a pandemic period, which places various forms of pressure on students [53]. It is also useful for instructors to try different methods to boost students' confidence and reduce their anxiety in learning and using the target language, such as praising them, giving clues, allowing more time to think, sharing language learning experiences, being empathetic, encouraging students to be positive and study hard and providing adequate, relevant and interesting materials [3,4,8,12,42,43]. Concurrently, students should be encouraged to try all means to improve their proficiency in the target language, have a proper attitude towards mistakes, face negative evaluation with courage, share learning experiences and feelings with peers and the instructor, be active and prepared for each class, increase exposure and access to the target language and adjust themselves psychologically to the pandemic context. Gradually, they will become less and less anxious and more and more confident in learning and using the target language in class.

Meanwhile, it is necessary for instructors to know their students' perceptual learning styles in order to engage them in a more effective learning and teaching process in an SL/FL classroom. Instructors are advised to prepare different teaching techniques and materials to attend to learners' diverse learning styles [15,16,19,23,25,58]. Auditory learners may prefer to hear written text material and engage in oral practice, visual students may need visual stimulation of bulletin boards and videos, individual learners may learn more efficiently by themselves and group learners may prefer to learn in groups [59]. A mismatch between the teaching style and a student's preferred learning style may interfere with learning, as "these mismatches are at the root of many learning difficulties" [59] (p. 50). Instructors are advised to prepare tasks and activities that require students to utilize different strategies, with instruction and materials presented in various forms. Students, as discussed in Nosratinia and Soleimannejad [17], will benefit from developing different perceptual learning styles and trying to diversify their learning techniques. As discussed in Ehrman [59], multi-style preference can help students make the most of the learning circumstance.

This study is one of the few on the relationship between English classroom anxiety and learning styles and their predicting effects on students' English achievement; it revealed interesting findings, thus contributing to the current literature. Theoretically, the findings help explain the relationship between FLA and learning styles and further pinpoint the important roles of FLA and learning styles in SL/FL learning. Pedagogically, the study sheds light on and offers specific suggestions for the teaching and learning of an SL/FL. These attest to the belief in the socio-interactionist theory that learning is shaped by interaction with internal and external factors [26,55]. With more empirical research, a path model

among learning styles, FLA and learning outcomes can be built. Nonetheless, some limitations existed in the study. First, this study was purely quantitative, though the validity and reliability of the results and interpretations could have been increased with qualitative data. Second, this study focused on English classroom anxiety and learning styles in relation to gender and discipline, and excluded other individual factors such as age and SL/FL proficiency level. Finally, as discussed in Dantas and Cunha [26], there are different theories on learning style and debates on its nature and role in learning, which deserve attention in future research. All these should be the focus of future research. Moreover, the results of the present study need to be confirmed in other empirical studies.

**Funding:** This research was funded by 2021 Top-Notch Students of Basic Disciplines Training Program 2.0 Project (Grant number 20222015).

**Institutional Review Board Statement:** The study was conducted in accordance with the Declaration of Helsinki, and approved by the Research Committee of the Department of Foreign Languages and Literatures, Tsinghua University.

**Informed Consent Statement:** Informed consent was obtained from all subjects involved in the study.

**Data Availability Statement:** Data will be available upon request.

**Conflicts of Interest:** The author declares no conflict of interest.

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
