# Peer review of "English Classroom Anxiety, Learning Style and English Achievement in Chinese University EFL Students"

_sustainability, doi:10.3390/su151813697_

Round 1
Reviewer 1 Report
Please separate the discussion and conclusion. The concluding remarks must incorporate.
result, and the discussion is not aligned, and need to fix this issue.
It's well written
Author Response
I am feel much indebted to the reviewer for the insightful comments and suggestions on my submission. All the comments and suggestions have been carefully considered and integrated into the revised manuscript, as detailed below and highlighted in the resubmission.
- Please separate the discussion and conclusion. The concluding remarks must incorporate result, and the discussion is not aligned, and need to fix this issue.
--yes, all done as suggested (pp.13-14).
- Comments on the Quality of English Language: It's well written
--Many thanks!
- The manuscript was carefully proofread several times to avoid problems of any kind.
Reviewer 2 Report
Dear authors
I reviewed the manuscript entitled “English Classroom Anxiety, Learning Style and English Achievement in Chinese University EFL Students”. This study is of great and important topic. In my opinion, it is greatly fall within the scope of journal. It should also be mentioned that the manuscript has addressed an issue that is my favorite subject area. Reading and evaluating different parts of the paper shows that the authors have made a lot of effort to carry out this research endeavor. Their efforts have resulted in important and ground-breaking conclusions that can certainly be used by different end-users including policy-makers, decision-makers, education development authorities, practitioners, extensionist, and so on. The authors have also used state of the art methodologies for analyzing their data. This makes their results and conclusions more reliable and rigorous. Therefore, I recommend this paper for publication. However, there are some points that should be addressed by the respected authors before consideration of the manuscript for publication in Sustainability. My main comments are as follows:
1. In the end of introduction, the main research sub-objectives should be mentioned.
2. Please highlight the most important originalities of the research in the end introduction.
3. The literature review section should be shortened.
4. The title of section 4 should be changed to “Methodology”. Then some sub-titles including Research Area, Population and Sampling, Reliability and Validity of the Research Tool, Data Analysis, and so on should be added to the methodology section.
5. Theoretical model (along with table 2 and figure 2) and research hypotheses (Section 3) should be moved to before methodology (After introduction).
6. Ho did you evaluate the convergent and divergent validity of the questions? This should be completely explained in the methodology section.
7. Please give some more details about the data collection process and data collecting team.
16. In conclusion section try to highlight the main contribution of your paper to the theory and practice.
In general, I believe that this manuscript can be accepted for publication in Sustainability after Major revisions.
Reviewer
Author Response
Please refer to the attached file "Responses to Reviewer 2"

Reviewer 3 Report
English Classroom Anxiety, Learning Style and English 2 Achievement in Chinese University EFL Students
General comments
- The linkage between Classroom Anxiety, Learning Style and English Achievement in EFl students sounds original. But the literature base does not help understanding why anxiety is a problem and how a linkage with learning styles might offer a solution of be a contribution.
- The linkage with the anxiety literature disappears altogether from the study.
- The learning styles literature has a long history and mirrors a fierce debate about it’s usefulness. This is neglected in this study. I expect that authors who adopt the concept have a strong grounding in the related literature and at least acknowledge the controversial evidence about the concept, its history and its fluctuating conceptualization. Moreover, as will be demonstrated, the author “sues” the literature in a selective way and even derives wrong conclusions from the literature actually being cited (this starts already with the first sentence in the article). This makes me as a reviewer suspicious as to the broader way literature has been “used” in this study.
- The authors mixes literature about learning styles with the cognitive style and working memory literature. This is not well elaborated. The linkage between working memory and cognitive/learning styles is very debatable and neglects the complex mechanisms detected in state-of-the-art literature about knowledge processing.
Specific comments
- P.1: I do not agree with the opening sentence :” The concept of learning styles is in many countries a textbook approach for tailoring pedagogical practice to individual differences in learning styles.”. This sentence neglects the strong controversies that can be found in the literature and especially the evidence-based teaching and learning approaches that have evolved far away from a simplistic conceptualization. Especially the idea that “tailoring” will work is the catch. Though it sounds attractive for teachers and educationists, this seems to be the key error in the conceptualization about learning styles: it is expected to be a stable characteristic of learners. Moreover, the author makes references to three publications at the end of this paragraph (one is not a publication but a project description), that explicitly state that the learning styles literature is not leading to a sound theoretical grounding. We simply cite here the authors actually being referred to in the article (Willingtham and colleagues, 2015): “There is reason to think that people view learning styles theories as broadly accurate, but, in fact, scientific support for these theories is lacking. We suggest that educators’ time and energy are better spent on other theories that might aid instruction.”.
- P1: The statement ‘Every individual learns in his or her own way, “ is hardly an eye opener and does not automatically justify that “learning styles” are at play. The cognitivist and neuropsychological research present a clear picture about the individual processing styles of learners that go far beyond a conceptualization in terms of ‘learning styles’. This is confirmed by the authors on p.2, but I do not agree that this justifies the adoption of the concept of learning styles. Indeed, the concept was popular in the 70ies till the 90ties, but since these days, the conceptualisation has hardly evolved.
- P.2 The learning styles literature being referred to is not up to date. Reference to Keefe and Reid are dated and neglect the large number of learning styles reviews published since. For instance: Aslaksen, K., & Lorås, H. (2019). Matching instruction with modality-specific learning style: Effects on immediate recall and working memory performance. Education Sciences, 9(1), 32. Or Rinekso, A. B. (2021). Pros and cons of learning style: An implication for English language teachers. Acuity: Journal of English Language Pedagogy, Literature and Culture, 6(1), 12-23.
Especially the underlying Aptitude Treatment Hypothesis is central to these critiques and the fact that learning styles are considered as “stable” learning characteristics. Why does the author not mention this literature? I do not take a particular stand in the discussion, but I expect that the literature base mirrors the ongoing debate.
- P.2: the review studies about the learning styles literature mentioned here introduce interesting avenues to further the discussion. But, to build on a 2008 article to ground the 2023 discussion does not seem justified; especially since directions for new research are derived from this; e.g., in terms of research design. Also, the review points at research involving ‘children’ . Is this the same research population as adopted in this study (higher education/university level)? Also, when referring to more recent studies, the most recent study is of 2018.
In addition, we observe how the authors interlinks studies set up in very different knowledge domains, involving different age groups and learning settings. It can hardly be expected that this might lead to a consistent useful theoretical and conceptual base.
- P.3 moves to another theoretical base to look at individual learning differences and looks at modalities used by learners to process information. The discussion about modality specific learning styles mirrors the same defects as mentioned earlier: dated references and neglecting state of the art discussions about modalities. The author enters here the domain of cognitive processing where information is often multi-modal. The consistent finding in the related literature is that the processing of the same information through multiple modalities will strengthen the construction of schema (see the cognitive theory of multimedia learning of Mayer and colleagues). To derive from this literature a foundation for a modality learning style has already been proven to be unsuccessful. Moreover, the author all the sudden introduces a new concept: cognitive style. Again, this is a controversial concept with a long history that is often linked to the learning style literature. The critical review studies about cognitive style are again neglected.
- On p.3, the author introduces Working memory and sees it as a candidate for cognitive ability. Why is this introduced at this position in the article? This is explained in the next paragraph. Suddenly, the individual differences that were linked to the learning styles and cognitive styles literature are linked to individual differences in working memory capacities. Of course there are differences in working memory capacities since they have to be developed, trained... And by looking at grounding literature about WM (such as Baddeley and Hitch), we know that modalities of information are being considered (mainly auditory and visual). But to link this literature to “styles” is a bold move. The literature mentioned here is not relevant to back this hypothesis. Moreover, modalities exceed a focus on auditory and visual stimuli. The state-of-the-art literature now points at kinesthetic, psychomotor, olfactorily, … and mixed modalities in the representation of knowledge that exceed the simplistic distinction between visual and auditory stimuli. It is therefore not surprising that the researcher does not find significant differences in the research being set up. The current literature base has moved already far ahead of the dated juxtaposition of two sensory representations.
- The juxtaposition of visual/auditory knowledge representation/processing preferences neglects the complex instructional approach that mixes multiple representation in English as a Foreign Language teaching approaches (EFL).
- P.4: Is a preference for a knowledge representation automatically the same as a learning style?
- P.4: moving ahead to the research approach being adopted, we criticize the reduction of EFL teaching and learning to “audio presentation and text.” This selective knowledge representation might hinder actual processing of the information.
- The prior knowledge about the topic “Vikings” is neglected in terms of the knowledge representations that are already available in long term memory on the base of earlier learning experiences. The researcher makes an attempt to check for prior knowledge differences (and via self assessment). But this neglects the representational base of the prior knowledge in long term memory.
- The number of participants is too low to be able to counter the impact of measurement errors (false negative/positive). The nature of the participants raises questions: 13 visual learners and 9 auditory learners. Why continuing with a skewed distribution in the sample? How can randomization will be adequate when only 9 auditory learner were assigned to one of two conditions?
- We criticize the adoption of parametric tests when working with such small (selective) samples (correlation, regression analysis).
- I criticize the use of the WAISS to assess working memory capacity. Much better and more recently developed and validates tests are available. No data as to the use with Norwegian subjects is being presented (what version, age group …).
- In the discussion section, there is adequate use of related literature t explain the findings. But some of this literature should have been used in the conceptual and theoretical base; e.g., Viewed from the perspective of the cognitive neurosciences, spoken and written words are processed in brain areas observed to correlate with language processing. And …. Again we criticize the uncritical use of dated literature that helps confirming one’s own opinion, but neglects state of the art research evidence that grounds the opposite position.
Author Response
Please refer to the attached file "Responses to Reviewer 3"

Round 2
Reviewer 1 Report
well revised
well revised
Author Response
Many thanks for your work and positive comments!
Reviewer 2 Report
Dear authors
I reviewed the responses and revisions and I believe that the manuscript can be accepted in present form. Thank you very much for addressing my comments.
Best,
Reviewer
Author Response
Many thanks for you work and positive comments!
Reviewer 3 Report
Dear authors,
Thank you for this revised version. I also thank you for understanding the nature of the rather negative review you received and that a thorough revision was necessary.
I will react to each of the comments in your rebuttal letter. You will understand that some issues remain to be dealt with.
General comments
- Linkage
1. Clarify link between anxiety and EFL: this is sufficiently tackled in the revised version.
2. Clarify link between learning styles and EFL anxiety: it is not sufficient to state that there is hardly literature about this. My main point was that the potential link is not explained from a theoretical point of view. What are the’ mechanisms that help explaining the importance and relevance of this linkage. This is a remark about the theoretical contribution of the research being presented.
3. To be able to talk about a “contribution” to the literature when it comes to the link between anxiety, EFL and learning styles, a theoretical explanation is needed. An empirical linkage is not sufficient since this does not explain why there might be an association, correlation, influence …. The authors build on the available literature that also does not present a theoretical explanation and hardly an empirical underpinning. I also refer to the results of the study of the authors on p.17: “The present study also reveals a significantly positive correlation between each learning style and students’ English achievement”. It is always satisfying to find a result that underpins hypotheses. But, and here I return to the above remark, how do you explain this relationship. What is the theory behind the finding? It does not suffice to refer to other authors who suggest or found this relationship. That is not a satisfactory explanation, and it does not contribute to theory development.
- The linkage with the anxiety literature disappears altogether from the study.
4. The authors state: “Because there is little research on learning styles and foreign language anxiety, no claim can be made on their relationship.” Above I stated why it is key to clarify from a theoretical point of the relationship you claim to exist. In my comment about “anxiety disappears from the manuscript” I gave an observation about a key relationship that is put forward at the start of their article but gets out of focus in their article. The claim about the relationship was not made by me, it is a claim of the authors themselves. I also refer to the actual results of the study. On p.12 the authors conclude “. These findings further support the negative effect of foreign language anxiety in SL/FL learning regardless of learning environments.” But I do not find an explanation WHY this relationship is supported. What is their theory about this relationship?
- The learning styles literature has a long history …
5. The authors state that the past and current debate is no concern of the research because they write about learning styles (LS) in second/foreign language learning. They refer to Reid’s theory. They implicitly state that Reid’s theory is a such not part of the debate. I do not agree with this statement. Reid builds on learning styles assumptions from the broader theoretical learning styles literature and as such is part of the debate. I expect at least reference to the debate and an acknowledgment about weaknesses of the LS approach to learning and instruction. Especially the ongoing debate about the dynamic or stable nature of learning styles is of concern here. In their section about LS, no critical comments are being considered. Immediately, the authors jump to an operationalization of the LS conception through instruments (e.g., PLSPQ).
- Cognitive styles and working memory
6. I agree that the comment about cognitive style might be less related to this article. But what I meant is that the link between learning styles and working memory is weakly explained.
Specific comments
- The concept of learning styles is in many countries a textbook approach for tailoring pedagogical practice to individual differences in learning styles.
7. I agree that this sentence is not part of the manuscript. I apologize for mixing up a text.
- The statement ‘Every individual learns in his or her own way
8. Thank you for acknowledging that the sentence implicitly suggests a link with learning styles. Especially the second sentence in this paragraph strongly makes this suggestion: “Consequently, learning styles have been increasingly researched in SL/FL learning in”. I recommend again to change the working to take a way this wrong suggestion.
- The learning styles literature being referred to is not up to date.
9. Thank you for adding recent studies. It was wise to add critical studies; such as the article of Rinesko (2021).
- Especially the underlying Aptitude Treatment Hypothesis is…
10. Indeed, you do not mention ATI, but you build explicitlty on ATI assumption because you suggest adapting the instructional approach to the learning style of learners. That is central to ATI.
11. The stability of learning styles is part of the ongoing debate. It does not suffice to select one author (article from 1995) that suggest that learning styles are stable indicators. This does not help underpinning the assumption . Empirical evidence should be at the center of the discussion and not referring to an “authority”.
- The review studies about the learning styles literature mentioned here introduce interesting avenues to further the discussion.
12. Thank you for adding more recent LS research.
- In addition, we observe how the authors interlinks studies set up in very different knowledge domains,
13. It is strange that the authors state in their reply: “If findings are consistent across genders, ages, disciplines and other individual characteristics, no more such research is needed then”. The available literature does not suggest such thing. I tried to indicate that there is no clear answer to the gender differences, disciplines … The inconsistency in the empirical literature does not underpin the theoretical statements.
- P.3 moves to another theoretical base to look at individual learning differences and looks at modalities used by learners to process information.
14. What I tried to clarify is that the use of the concept of modality opens a different theoretical angle in the learning styles literature. It is not because the learning styles literature neglects the up-to-date cognitive processing literature about modalities that this literature is not relevant. If the authors do not want to make this linkage, it is their choice. I recommend reading e.g.,
Cuevas, J., & Dawson, B. L. (2018). A test of two alternative cognitive processing models: Learning styles and dual coding. Theory and Research in Education, 16(1), 40-64.
Lodge, J. M., Hansen, L., & Cottrell, D. (2016). Modality preference and learning style theories: rethinking the role of sensory modality in learning. Learning: Research and Practice, 2(1), 4-17.
15. I admit that I - as a reviewer - introduced here the concept of cognitive style; which I am most willing to remove from the comment.
All the remaining observations of the authors are correct. The review text contained on a last page a section of another article review. I apologize for this.
Please check the typos in your revised text. This is easy to be dealt with.
Author Response
Many thanks for your work and comments!
